# Understanding the uptake of new hip replacement implants in the UK: a cohort study using data from the National Joint Registry for England and Wales

Chris M Penfold [ID],[1,2] Ashley W Blom,[1,2] Adrian Sayers [ID],[1,3] J Mark Wilkinson,[4,5] Linda Hunt,[1] Andrew Judge,[1,2] Michael R Whitehouse [ID] [1,2]

For numbered affiliations see end of article.

**Correspondence to**
Dr Chris M Penfold;
Chris.Penfold@bristol.ac.uk

## ABSTRACT

**Objectives** Primary: describe uptake of new implant components (femoral stem or acetabular cup/shell) for total hip replacements (THRs) in the National Joint Registry for England and Wales (NJR). Secondary: compare the characteristics of: (a) surgeons and (b) patients who used/received new rather than established components.

**Design** Cohort of 618 393 primary THRs performed for osteoarthritis (±other indications) by 4979 surgeons between 2008 and 2017 in England and Wales from the NJR. We described the uptake of new (first recorded use >2008, used within 5 years) stems/cups, and variation in uptake by surgeons (primary objectives). We explored surgeon-level and patient-level factors associated with use/receipt of new components with logistic regression models (secondary objectives).

**Outcomes** Primary outcomes: total number of new cups/stems, proportion of operations using new versus established components. Secondary outcomes: odds of: (a) a surgeon using a new cup/stem in a calendar-year, (b) a patient receiving a new rather than established cup/stem.

**Results** Sixty-eight new cups and 72 new stems were used in 47 606 primary THRs (7.7%) by 2005 surgeons (40.3%) 2008–2017. Surgeons used a median of one new stem and cup (25%–75%=1–2 both, max=10 cups, max=8 stems). Surgeons performed a median total of 22 THRs (25%–75%=5–124, range=1–3938) in the period 2008–2017. Surgeons used new stems in a median of 5.0% (25%–75%=1.3%–16.1%) and new cups in a median of 9.4% (25%–75%=2.8%–26.7%) of their THRs. Patients aged <55 years old versus those 55–80 had higher odds of receiving a new rather than established stem (OR=1.83, 95% CI=1.73–1.93) and cup (OR=1.31, 95% CI=1.25–1.37). Women had lower odds of receiving a new stem (OR=0.87, 95% CI=0.84–0.90), higher odds of receiving a new cup (OR=1.06, 95% CI=1.03–1.09).

**Conclusions** Large numbers of new THR components have been introduced in the NJR since 2008. 40% of surgeons have tried new components, with wide variation in how many types and frequency they have been used.

### Strengths and limitations of this study

- ► This study provides a nationally representative description of the uptake of new implant components for total hip replacements in England and Wales.
- ► This is the first study to describe the variation in uptake of new components by surgeons, and surgeon characteristics which may be associated with the use of new components.
- ► Although implant component brand names were checked by the authors, some components may have been reclassified or we may still have misclassified some components as either new or established, but the introduction of unique device identifiers should remove this problem in future.
- ► The surgeon assigned as lead operating surgeon in the National Joint Registry for England and Wales may not be correct, although consistency between our sensitivity and primary analyses indicate that this is unlikely to have substantially affected our findings.
- ► Hospital-level or regional variation in suppliers may be important factors affecting implant uptake, but these were beyond the scope of this study.

### BACKGROUND

Total hip replacements (THRs) are mainly performed to treat pain and functional limitation due to osteoarthritis (OA).[1] It is a highly successful surgical procedure with typical 10-year revision rates<5%,[2] the current National Institute for Health and Care Excellence benchmark.[3] However, younger patients are more likely to require revision surgery; the lifetime revision risk for men having a THR in their 50 s is ~35% compared with 5% in their 70s.[4] Such patients may benefit the most from developments in THR that lead to reduced revision rates or improved outcomes. However, they may also be affected

for the longest time if these developments lead to poorer outcomes.

Some new implant designs intended to benefit these more active and/or younger patients have been high-profile failures, for example metal-on-metal THRs[5] including the Articular Surface Replacement (ASR) prostheses in particular.[6] Many new implants, including the ASR,[7] were introduced with minimal supporting evidence of their effectiveness[8] and may offer at best no improvement over pre-existing components.[9] An influential agenda for surgery research (IDEAL) was developed, providing a framework for future investigations into surgical innovations, which recommended the phased introduction of new medical devices.[10] The rapid uptake of ASR hip replacements before the publication of supporting evidence bypassed IDEAL Stages 2a ('Development') and 2b ('Early dispersion and exploration'). Instead, long-term monitoring was relied on to monitor outcomes (Stage 4).[7] It is not clear whether the uptake of newer implants has also been rapid.

There is wide variation between and within regions in the use of common surgical procedures, which are only explained to a small degree by differing patient demands and diagnostic practices.[11] The large number of different components used in primary THRs (127 femoral stems and 105 acetabular cups recorded in the National Joint Registry for England, Wales, Northern Ireland and the Isle of Man (NJR) in 2016)[12] may be an important source of variation. Many registries describe the volume of different implant components used annually but not the variation in uptake of new implants between surgeons or which patients receive them. More research is needed to understand and reduce avoidable variation in outcomes created by differences in surgical activity.

We aimed to:
1. Describe the uptake of new implants for THRs in the NJR.
2. Describe how this uptake varies by surgeons.
3. Compare surgeons who use new compared with established components.
4. Compare patients who receive new compared with established components.

## METHODS
### Data source
The NJR was established in 2003.[2] Data entry for Northern Ireland and the Isle of Man did not commence until 2013 and 2015, respectively; therefore, they are excluded from this analysis. Key markers of NJR data quality were high and stable from 2008 onwards.[13]

### Study sample
We included the cohort of patients who received a primary THR for OA (±other indications) between 1 January 2008 and 26 February 2017. We used NJR data from 2003 onwards to calculate the date each implant component was first used and the total number of implantations.

We excluded people who had not given consent for recording of personal details, where the brand of their acetabular or femoral components was uncertain, and those who received a resurfacing rather than stemmed THR. Resurfacing THRs were excluded since patients who receive these are a very different demographic from those receiving stemmed THRs (significantly younger and more likely to be male), and the annual volume is very low (~550 in 2017) and decreasing.[14]

### Patient involvement
This study was designed and undertaken without patient involvement.

### Definition of new and established implant components
We identified the implant component brand from component labels recorded in the NJR. We used the earliest recorded use by any surgeon in the NJR of each femoral (stem) or acetabular (cup or shell) component to define an implant component's start date. We classified implant components with a start date between the beginning of NJR data collection (2003) and the end of 2007 as 'established'. This allowed implant components which were in use before the NJR started but which may have only been used occasionally to be recorded in the NJR and classified appropriately as 'established'. NJR data quality was also high and stable from 2008 onwards. Implant components with a start date on or after 1 January 2008 and which were used within 5 years of this start date were classified as 'new'. Those used later than 5 years after their start date were classified as 'established'.

### Surgeon uptake of new implant components
All surgeons with operations recorded in the NJR are assigned an anonymised identifier, and their role in the operation ('consultant in charge' or 'operating') is recorded. We summarised each operating surgeon's activity across each calendar-year in which they performed ≥1 THR. We considered five potential surgeon-level factors which may be associated with use of a new component in a calendar-year: total volume of THRs performed in that year, proportion of those THRs performed on patients<55 years old (<10% and≥10%), source of funding for THRs ('100% National Health Service (NHS) funded' or 'some or all privately funded'), proportion of THRs performed on patients with an American Society of Anaesthesiologists (ASA) grade III–V (<25% and≥25%), and the range of different stem-cup combinations used in that calendar-year ('≤3', '4–6', '7–10' and '>10'). Surgeons who performed ≥10% of their THRs on patients aged <55 years old and those who performed ≥25% of their THRs on patients with ASA III–V were in approximately the upper quartile of these distributions.

### Patients receiving new implant components
We used date of surgery to order patients within implant components and within surgeons. We categorised patients according to whether the component they received was new or established. We considered five potential

patient-level factors which may be associated with their receipt of new components: age at the time of THR (<55, 55–80 and 80+ years), gender, body mass index (BMI), ASA grade and NHS or private funding. We selected these categories for age to reflect patients who were having a primary THR at a relatively young or relatively old age, the median age at the time of primary THR was 69 years (25%–75% 61–76 years).[14]

## Statistical analyses

We described the use of unique stems and cups in primary THRs performed since 1 January 2008, the cumulative use of new components in patients and the count of surgeons who used new components. We also described the total number of all and new cups, stems and combinations.

### Surgeon-level factors

In analyses of surgeon-level and patient-level factors associated with use of or receipt of new implants we included only those people with complete exposure and outcome data for the surgeon-level and patient-level analysis models (ie, complete case analysis). We assumed that data were missing at random but did not use multiple imputation to account for these missing data since there were no variables in the NJR dataset which were not already in our regression models and which may have carried information about the missing data (particularly BMI).

Our outcome was whether a surgeon used a new component at least once for a THR in a calendar-year (stems and cups analysed separately), unit of analysis was surgeon calendar-years and exposure variables were those surgeon-level factors defined previously. We used multivariable logistic regression models, accounting for the clustering of calendar-years within surgeons.

### Patient-level factors

Our outcome was whether a patient received a new rather than established component (stems and cups analysed separately), unit of analysis was patients and exposure variables were those patient-level factors defined previously. Patient-level factors were included in multivariable mixed-effects logistic regression models, with patients nested within surgeons.

### Sensitivity analyses

We conducted three sensitivity analyses. To determine whether the lack of variability in patients operated on by low volume surgeons affected our results we repeated our surgeon-level analysis excluding calendar-years for surgeons in which they performed <10 THRs. We also considered that the choice of component was made by the consultant in-charge rather than the operating surgeon (the consultant in-charge was not the operating surgeon for ~16% of THRs). We repeated our surgeon-level analysis by consultant in-charge and repeated our patient-level analysis with patients clustered within consultant in-charge.

In order to determine the extent to which patients with complete data for all exposures and outcome variables

differed from those missing some exposure data (mainly BMI) we compared these groups using chi-square tests. We also repeated our patient-level analyses for those patients with complete data for all exposure variables (including BMI) but excluding BMI from the model, and for those with complete data for all exposure variable (excluding BMI).

All analyses were performed using Stata V.15 (StataCorp).

## RESULTS

### Overall use of implant components

Between 1 January 2008 and 26 February 2017, 618 393 primary THRs were performed for OA in England and Wales and recorded in the NJR, corresponding to 23 887 calendar-years in which surgeons performed ≥1 THR. The mean age of the patients was 68.5 years (SD=11.1 years), 60.7% were female, their ASA grades were I: 14.2%, II: 69.9%, III: 15.5% and IV/V: 0.5%. Twenty-three per cent had a normal/underweight BMI, 39.6% were overweight and 37.6% obese. THRs were performed by 4979 surgeons using 189 different stems, 187 cups and 2026 stem-cup combinations. Surgeons used a median of three different stems (25%–75%=2–5, max=21), four cups (25%–75%=2–7, max=27) and five combinations (25%–75%=2–9, max=60). They performed a median total of 22 THRs between 2008 and 2017 (25%–75%=5–124, range 1–3938), although this includes surgeons who started part way through this period, retired or changed their practice. Excluding calendar-years in which a surgeon performed no THRs, the median number of THRs surgeons performed per year was 11 (25%–75%=3–35, range 1–584) and in 47% of surgeon calendar-years (11 164 of 23,887) surgeons performed <10 THRs.

### Use of new implant components

During this period 68 new cups (47 uncemented, online supplementary table S1) and 72 new stems (51 uncemented, online supplementary table S2) were first used. The rate of introduction of new cups and stems remained stable (~16 new components/year, online supplementary figure S1). Eight per cent (n=47 606) of THRs performed used a new stem, cup, or combination. Forty per cent (n=2005) of surgeons who performed a THR in this period used at least one new implant component.

New cups were used in 5.8% (n=35 885) THRs performed by 34.1% (n=1699) surgeons (online supplementary table S1), new stems in 2.9% (n=18 159) THRs by 22.3% (n=1111) surgeons (online supplementary table S2) and new combinations in 1.0% (n=6438) THRs by 8.7% (n=433) surgeons. Most new cups (n=19 775, 55.1%) and almost all new stems (n=15 361, 84.6%) were uncemented. The median number of new stems, cups and combinations used by surgeons was one (25%–75%=1–2, cups max=10, stems max=8 and combinations max=9; online supplementary table S3 - S5). The median THRs performed using new stems was

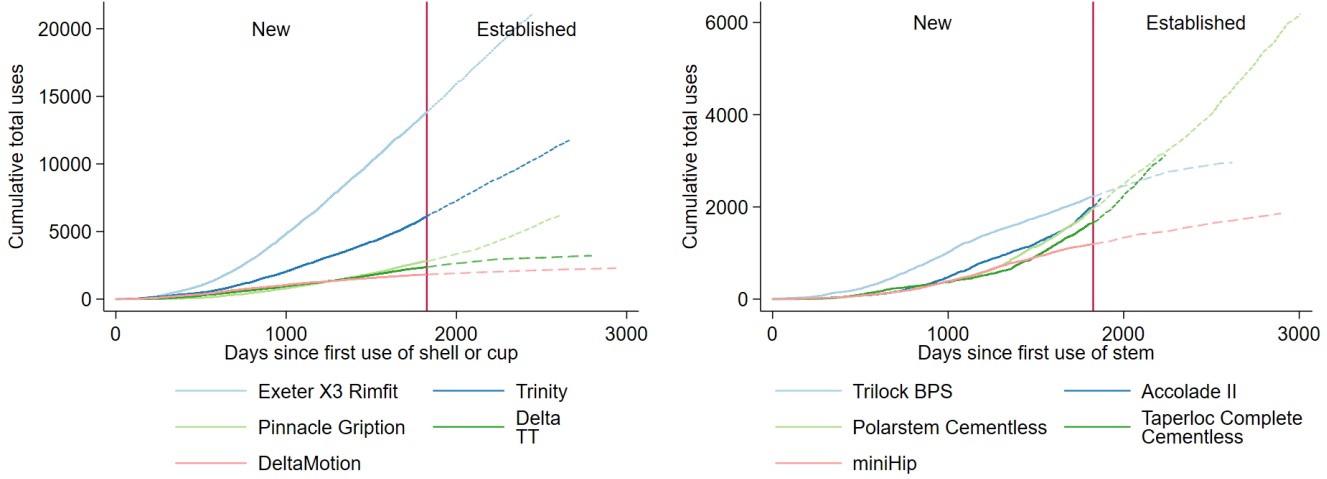

**Figure 1** Cumulative total use of the top five new stems and cups/shells by days since they were introduced.

three (25%–75%=1–11, max=637) and new cups was three (25%–75%=1–14, max=867). The median proportion of a surgeon's THRs performed using new stems was 3.4% (25%–75%=1.0%–10.6%), new cups 6.3% (25%–75%=2.0%–18.8%) and new combinations 2.4% (25%–75%=0.7%–9.1%).

The five most frequently implanted new stems were used in 9049 THRs (49.8% of THRs using a new stem, online supplementary table S2). The five most frequently implanted new cups were used in 26 962 THRs (75.1% of THRs using a new cup, online supplementary table S1). Uptake of the two most popular new cups was rapid (5000 uses of Exeter X3 Rimfit 1016 days, 5000 uses of Trinity 1651 days after first use, figure 1) but was slower for new stems (2000 uses of Polarstem Cementless 1670 days, figure 1). Conversely, a third of the new stems and cups (n=26/72 new stems, n=25/69 new cups) have been used in ≤10 THRs, and most of these have been used in ≤5 THRs (n=22 stems, n=20 cups).

**Surgeon-level and patient-level factors associated with new implant components**

Our complete case analysis included 431 955 out of a possible 618 393 THRs (69.8%) and 20 410 out of a possible 23 887 surgeon calendar-years (85.4%, online supplementary figure S2). We were missing patient-level data for BMI (n=1 86 308, 30.1%) and source of funding (n=1514, 0.2%). The characteristics of the subset of patients with complete data are shown in online supplementary table S6. There were minor differences between people with complete data and those with incomplete data (online supplementary table S6). Compared with people with incomplete data, a smaller proportion of people with complete data were aged ≥80 years old (14.8% vs 16.4%), female (60.3% vs 61.6%) and had their operation funded through the NHS (86.9% vs 89.4%).

**Characteristics of surgeons using new implant components**

Multivariable associations between surgeon-level factors and their use of new components in a calendar-year were consistent between stems and cups (table 1, unadjusted online supplementary table S7). Surgeons who treated more younger patients had 47% higher odds of using a new stem (OR=1.47, 95% CI 1.30 to 1.66, p<0.001) and 39% higher odds of using a new cup (OR=1.39, 95% CI 1.25 to 1.53, p<0.001) in a calendar-year. Those who performed more THRs/year had 6% higher odds of using new cups (OR=1.06, 95% CI 1.04 to 1.08, p<0.001) and 2% higher odds of using new stems (OR=1.02, 95% CI 1.00 to 1.05, p=0.206), although the CI crossed the null. Private funding was associated with 23% increased odds of using new stems (OR=1.23, 95% CI 1.05 to 1.43, p=0.010) and weakly associated with 9% increased odds of using new cups (OR=1.09, 95% CI 0.96 to 1.23, p=0.187) with confidence intervals crossing the null value. Use of more stem-cup combinations was strongly associated with increased use of new components (ORs for '>10' vs '≤3' combinations: 27.4 and 13.3 for stems and cups, respectively; p values<0.001). Proportion of patients with ASA grades III–IV was weakly associated with 12% higher odds of using new cups (OR=1.12, 95% CI 1.01 to 1.25, p=0.034) but not with using new stems (OR=1.01, 95% CI 0.89 to 1.16, p=0.843).

**Characteristics of patients receiving new implant components**

A higher proportion of recipients of new compared with established implant components were aged <55 years old (10.5% established vs 21.3% new stems; 10.5% established vs 14.8% new cups; table 2), although the main recipients of all components were aged 55–80 years. Fifteen per cent of recipients of established stems (15.0%) were ≥80 years old compared with 8.3% of recipients of new stems, but there was little difference in the proportion of older recipients of established (14.9%) and new (13.3%) cups.

**Table 1** Results from multivariable logistic regression models showing the association between surgeon-level factors and use of new stems and cups

| Exposure | Stems | | | | | Cups | | | | |
|---|---|---|---|---|---|---|---|---|---|---|
| | Established (n=18 404)* | New (n=2006)* | OR† | (95% CI) | P value† | Established (n=17 167)* | New (n=3243)* | OR† | (95% CI) | P value† |
| **Proportion of THRs performed on patients <55 years old** | | | | | | | | | | |
| <10% (ref) | 13 088 (71.1%) | 940 (46.9%) | 1 | – | – | 12 346 (71.9%) | 1682 (51.9%) | 1 | – | – |
| ≥10% | 5316 (28.9%) | 1066 (53.1%) | 1.47 | 1.30 to 1.66 | <0.001 | 4821 (28.1%) | 1561 (48.1%) | 1.39 | 1.25 to 1.53 | <0.001 |
| **Number of THRs performed in calendar year‡ (per 10 additional cases)** | 8 (2, 24) | 32 (12, 61) | 1.02 | 0.99 to 1.04 | 0.206 | 7 (2, 22) | 28 (10, 56) | 1.06 | 1.04 to 1.08 | <0.001 |
| **Proportion of THRs funded privately** | | | | | | | | | | |
| 100% NHS funded (ref) | 12 922 (70.2%) | 966 (48.2%) | 1 | – | – | 12 159 (70.8%) | 1729 (53.3%) | 1 | – | – |
| Some or all funded privately | 5482 (29.8%) | 1040 (51.8%) | 1.23 | 1.05 to 1.43 | 0.010 | 5008 (29.2%) | 1514 (46.7%) | 1.09 | 0.96 to 1.23 | 0.187 |
| **Number of stem–cup combinations used in calendar year** | | | | | | | | | | |
| ≤3 (ref) | 14 259 (77.5%) | 589 (29.4%) | 1 | – | – | 13 599 (79.2%) | 1249 (38.5%) | 1 | – | – |
| 4–6 | 3394 (18.4%) | 822 (41.0%) | 4.91 | 4.25 to 5.67 | <0.001 | 2937 (17.1%) | 1279 (39.4%) | 3.77 | 3.36 to 4.23 | <0.001 |
| 7–10 | 675 (3.7%) | 468 (23.3%) | 12.5 | 10.1 to 15.4 | <0.001 | 568 (3.3%) | 575 (17.7%) | 7.21 | 6.01 to 8.67 | <0.001 |
| >10 | 76 (0.4%) | 127 (6.3%) | 27.4 | 17.9 to 41.7 | <0.001 | 63 (0.4%) | 140 (4.3%) | 13.3 | 9.20 to 19.2 | <0.001 |
| **Proportion of THRs performed on patients with ASA grade III–V** | | | | | | | | | | |
| <25% (ref) | 13 244 (72.0%) | 1554 (77.5%) | 1 | – | – | 12 362 (72.0%) | 2436 (75.1%) | 1 | – | – |
| ≥25% | 5160 (28.0%) | 452 (22.5%) | 1.01 | 0.89 to 1.16 | 0.843 | 4805 (28.0%) | 807 (24.9%) | 1.12 | 1.01 to 1.25 | 0.034 |

*Proportions displayed are based on surgeon-calendar years.
†ORs, 95% CI and p values are from logistic regression models adjusted for all exposure variables.
‡Median (lower to upper quartile).
ASA, American Society of Anaesthesiologists; NHS, National Health Service; THR, total hip replacement.

**Table 2** Results from multivariable mixed-effects regression models (patients nested within surgeons) of age, gender, categorised BMI, ASA grade and source of funding on stem age and cup age, with category proportions

| | Stems | | | | | Cups | | | | |
|---|---|---|---|---|---|---|---|---|---|---|
| | Established (n=4 18 831) | New (n=13 124) | OR* | (95% CI) | P value | Established (n=4 06 072) | New (n=25 883) | OR* | (95% CI) | P value |
| **Age** | | | | | | | | | | |
| <55 years old | 43 780 (10.5%) | 2793 (21.3%) | 1.83 | 1.73 to 1.93 | <0.001 | 42 752 (10.5%) | 3821 (14.8%) | 1.31 | 1.25 to 1.37 | <0.001 |
| 55–80 (ref) | 312 205 (74.5%) | 9246 (70.5%) | 1 | – | – | 302 823 (74.6%) | 18 628 (72.0%) | 1 | – | – |
| ≥80 years old | 62 846 (15.0%) | 1085 (8.3%) | 0.60 | 0.56 to 0.64 | <0.001 | 60 497 (14.9%) | 3434 (13.3%) | 0.91 | 0.87 to 0.95 | <0.001 |
| **Gender** | | | | | | | | | | |
| Male (ref) | 165 607 (39.5%) | 5768 (44.0%) | 1 | – | – | 161 248 (39.7%) | 10 127 (39.1%) | 1 | – | – |
| Female | 253 224 (60.5%) | 7356 (56.0%) | 0.87 | 0.84 to 0.90 | <0.001 | 244 824 (60.3%) | 15 756 (60.9%) | 1.06 | 1.03 to 1.09 | <0.001 |
| **BMI** | | | | | | | | | | |
| Underweight and normal (ref.) | 95 306 (22.8%) | 2911 (22.2%) | 1 | – | – | 91 863 (22.6%) | 6354 (24.5%) | 1 | – | – |
| Overweight | 165 849 (39.6%) | 5138 (39.1%) | 1.02 | 0.97 to 1.08 | 0.373 | 160 834 (39.6%) | 10 153 (39.2%) | 0.95 | 0.91 to 0.99 | 0.007 |
| Class I obese | 105 670 (25.2%) | 3391 (25.8%) | 1.06 | 1.00 to 1.12 | 0.067 | 102 781 (25.3%) | 6280 (24.3%) | 0.93 | 0.90 to 0.97 | 0.001 |
| Class II obese | 38 995 (9.3%) | 1276 (9.7%) | 1.10 | 1.02 to 1.19 | 0.011 | 37 977 (9.4%) | 2294 (8.9%) | 0.94 | 0.89 to 1.00 | 0.042 |
| Class III obese | 13 011 (3.1%) | 408 (3.1%) | 0.99 | 0.87 to 1.11 | 0.808 | 12 617 (3.1%) | 802 (3.1%) | 0.94 | 0.86 to 1.02 | 0.135 |
| **ASA grade** | | | | | | | | | | |
| I (ref) | 60 022 (14.3%) | 2661 (20.3%) | 1 | – | – | 58 265 (14.3%) | 4418 (17.1%) | 1 | – | – |
| II | 293 142 (70.0%) | 8940 (68.1%) | 0.81 | 0.77 to 0.86 | <0.001 | 284 437 (70.0%) | 17 645 (68.2%) | 0.98 | 0.95 to 1.03 | 0.461 |
| III | 63 904 (15.3%) | 1482 (11.3%) | 0.66 | 0.61 to 0.72 | <0.001 | 61 681 (15.2%) | 3705 (14.3%) | 1.00 | 0.94 to 1.05 | 0.935 |
| IV+V | 1763 (0.4%) | 41 (0.3%) | 0.64 | 0.46 to 0.90 | 0.010 | 1689 (0.4%) | 115 (0.4%) | 1.02 | 0.82 to 1.26 | 0.881 |
| **Source of funding** | | | | | | | | | | |

Continued

**Table 2** Continued

| | Stems | | | | | Cups | | | | |
|---|---|---|---|---|---|---|---|---|---|---|
| | Established (n=4 18 831) | New (n=13 124) | OR* | (95% CI) | P value | Established (n=4 06 072) | New (n=25 883) | OR* | (95% CI) | P value |
| NHS | 364 928 (87.1%) | 10 553 (80.4%) | 1 | – | – | 354 642 (87.3%) | 20 839 (80.5%) | 1 | – | – |
| Private | 53 903 (12.9%) | 2571 (19.6%) | 1.02 | 0.95 to 1.08 | 0.642 | 51 430 (12.7%) | 5044 (19.5%) | 1.09 | 1.04 to 1.14 | <0.001 |

*ORs, 95% CI and p values are from mixed-effects logistic regression models adjusted for all exposure variables.

ASA, American Society of Anaesthesiologists; BMI, body mass index; NHS, National Health Service.

Across all components and component age, women were the main recipients of THRs. There was no difference in BMI between recipients of established and new stems or cups. A higher proportion of recipients of new components had ASA grade I (20.3% new vs 14.3% established stems; 17.1% new vs 14.3% established cups). A higher proportion of people with privately funded THRs had new components (stems: 19.6% new vs 12.9% established; cups: 19.5% new vs 12.7% established).

Multivariable mixed effects logistic regression models (table 2, unadjusted online supplementary table S8) found that patients<55 years old, compared with those 55–80, had 83% and 31% higher odds of receiving a new rather than established stem (OR=1.83, 95% CI 1.73 to 1.93, p<0.001) and cup (OR=1.31, 95% CI 1.25 to 1.37, p<0.001). Women had 13% lower odds than men of receiving a new stem (OR=0.87, 95% CI 0.84 to 0.90, p<0.001), but 6% higher odds of receiving a new cup (OR=1.06, 95% CI 1.03 to 1.09, p<0.001). There was weak evidence that people with higher BMI had 10% higher odds of receiving a new stem (OR for underweight/normal vs class II obese=1.10, 95% CI 1.02 to 1.19, p=0.011) and weak evidence for the converse association between BMI and receiving a new cup (eg, OR for underweight/normal vs class II obese=0.94, 95% CI 0.89 to 1.00, p=0.042). Higher ASA grade was associated with 36% lower odds of receiving new stems (OR for ASA grades 'IV+V' vs 'I'=0.64, 95% CI 0.46 to 0.90, p=0.010), but was not associated with receiving new cups (OR for ASA grades 'IV+V' vs 'I'=1.02, 95% CI 0.82 to 1.26, p=0.881). Patients with private versus NHS funding had 9% higher odds of receiving new cups (OR=1.09, 95% CI 1.04 to 1.14, p<0.001), but there was no association between source of funding and receiving new stems (OR=1.02, 95% CI 0.95 to 1.08, p=0.642).

### Sensitivity analyses

Results of our first sensitivity analyses (excluding calendar-years for surgeons with <10 THRs) differed only minimally from our primary analyses (online supplementary table S9), indicating that our results were not biased by low-volume surgeons. In our second sensitivity analyses ('consultant in-charge' as the clustering variable) associations between source of funding and receipt of new stem/cup were stronger, otherwise they differed only minimally differed from our primary analyses (online supplementary tables S10, S11). Our comparison of regression models without BMI as an exposure, with complete cases as defined previously (n=431 955) and complete cases defined without BMI (n=616 879) found only minor differences. This suggests that associations between the exposures and outcomes for the population missing BMI differ only slightly from the population with BMI.

### DISCUSSION

Sixty-eight new cups and 72 new stems were first used in THRs in the NJR for OA between 2008 and 2017. Most

THRs used components introduced before 2008 but 8% used a new stem or cup. Uptake of some new implant components was very rapid. Conversely, uptake of a third of new components has been slow. Most surgeons used a maximum total of seven different cups or stems, of which one or two were new components. A small number of surgeons used a wide variety of different components, including new stems, cups and combinations.

Strengths of our study include the use of the NJR dataset, the largest arthroplasty register with comprehensive data capture (>95% in the period studied). This is the first to describe the variation in factors associated with uptake of new implant components by surgeons and receipt of new components by patients. Our study has several weaknesses. We classified a component as new based on the first record of a brand name in the NJR, but this does not exclude the possibility that a component was introduced earlier to other markets outside the UK. Furthermore, new components may constitute procedures not uploaded to the NJR (missing primary THRs estimated <5%). Also, some of these components may be minor modifications or a rebadged/renamed version of an existing component and some may also cover successive versions of a component. The correct operating surgeon may not be assigned to every operation. The extent to which this applies is unknown but may result in inaccurate estimates of surgeon-level associations, although our sensitivity analyses indicate that this is unlikely. The associations we have reported may be confounded by unmeasured factors (residual confounding) and in the absence of pre-existing literature on the uptake of new implants the findings from the regression models should be considered exploratory. We were missing BMI data for some people and elected not to use multiple imputation to account for these missing data, however our sensitivity analyses suggest that people with BMI data did not differ substantially from those without BMI across our other measures. Finally, we did not have data on hospital-level factors or regional variation in suppliers in our analyses which may be drivers of selection.[15]

Approximately 16 new implant components/year (stems and cups) were introduced in the NJR between 2008 and 2017. Comparisons with Australia (34 implant components/year 2003–2008)[16] and Finland (2–4 components/year 1980–2013)[17] suggest that this rate is not unusual, but that there is large variation internationally. The rapid uptake of some new components indicates that phased introduction, as recommended in the IDEAL Framework and others,[18] is unlikely to be happening. It is unclear whether 16 new implant components/year is of itself a good or bad thing. However, a healthcare system which supported a graduated introduction of new components, where the use of new components is restricted to specialised centres,[18] would provide a natural limit on the rate of introduction of new components until satisfactory and robust evidence is generated to support their more widespread use. Conversely, a third of new implant components have not yet accrued more than ten uses.

Postmarket surveillance of THRs, due to their longevity, performs a safety monitoring role which cannot easily be replaced by preapproval clinical data. Since the statistical methods are not applicable to components used in small numbers collaboration between international arthroplasty registries may allow more effective monitoring for low-volume components.

Over half of surgeons in our study used ≤5 different stems, cups or combinations, similar to a median of two different implant brands reported by surgeons in the USA in 1997.[19] The volume of THRs performed by surgeons using new components was often low (median ≤3 THRs with new components vs median 22 THRs in total), but the proportion of their THRs using any new components varied from 1% (lower quartile) to 19% (upper quartile). Surgeons who use a wider range of prosthesis combinations in THRs may have higher revision rates[20] and early THRs performed after switching implants may have a higher revision risk (also known as 'learning-curve').[21] While this suggests that surgeons should rely on a narrow range of implant components and rarely switch, a phased introduction of new implant designs, as is done in Sweden, may mitigate the learning-curve effect.[22] Since there are no contemporary comparisons of the range of implant components surgeons use and their relative volumes, it is unclear whether the between-surgeon variation we have reported may be associated with worse implant survival and warrants further research.

We found that newer components were being used in patients likely to be more active (ie, younger and/or male patients). There has been increasing evidence that uncemented implants, particularly stems, should not be used in older patients, but some uncertainty remains about their use in young patients (especially uncemented cups).[23–25] Since the majority of new cups and stems are uncemented, the decision to use these implant components in younger patients may increase the already high lifetime risk of revision surgery for these patients. Associations between BMI or ASA grade and receipt or use of new components were inconsistent between stems and cups and did not provide clear support for the use of new implant components in patients likely to be more active (ie, lower BMI and ASA grades). It may be of interest to further investigate the implant component choices made for patients with higher BMI or ASA grades.

The most comparable previous work used NJR data to explore patient-level and hospital-level determinants that patients receive uncemented versus cemented implants.[15] Uncemented components were less likely to be used in women and older patients, and hospitals treating older patients were less likely to use them. Our results indicate that surgeons who treat a higher proportion of younger patients are more likely to use newer components. Our most marked finding, that surgeons who used a wide variety of stem-cup combinations (either established or new) were much more likely to try a new component, may be somewhat self-evident but suggests that there may be a subset of surgeons who change components more

quickly than their peers. Whether this behaviour, along-side the previously discussed learning-curve, is related to outcomes of THRs is currently unclear.

Proposals for how new implant components should be introduced have been made previously, largely focused on phased introduction through high-volume centres and surgeons, and reliance on registries for long-term monitoring. It seems unlikely that 16 new THR implant components/year, as we found in our study, could be sustained through such an approach. Alongside the potential benefits of phased introduction discussed elsewhere, this approach would probably reduce the number of implant components used only in very low numbers. Since these are not monitored in the same manner as higher volume components this would probably be a good thing for patients, providing implant components intended for use in specialist cases are not adversely affected.

Further research could build on the findings of this study in several ways. Extending our analysis of surgeon-level factors associated with uptake of new components to include factors associated with risk of revision after THR would be valuable to surgeons and patients. Specifically, the 'learning curve' associated with changing implants and the complex relationship between surgeon's volume and outcomes. In addition, widening our study to cover hospital-level factors or regional variation in suppliers may highlight other drivers of selection.

## CONCLUSIONS

A large number of new THR implant components have been introduced into use in the NJR since 2008. The majority of THRs performed since 2008 used components which have been in use for a long time, but a large number of surgeons have tried new components, with wide variation in how many types and how often they have been used. The impact of this variation on patient outcomes is currently unclear. New rather than established implant components are more likely to be used in patients who are younger and/or male, although whether this will reduce the high lifetime risk of revision for this population is unclear.

**Author affiliations**
[1]Musculoskeletal Research Unit, Translational Health Sciences, Bristol Medical School, University of Bristol, Bristol, UK
[2]National Institute for Health Research Bristol Biomedical Research Centre, University Hospitals Bristol NHS Foundation Trust and University of Bristol, Bristol, UK
[3]Population Health Sciences, Bristol Medical School, University of Bristol, Bristol, UK
[4]Metabolic Bone Unit, Sorby Wing, Northern General Hospital, Sheffield, UK
[5]Department of Oncology and Metabolism and The Mellanby Centre for Bone Research, University of Sheffield, Sheffield, UK

**Acknowledgements** This study was funded by the NIHR Biomedical Research Centre at University Hospitals Bristol NHS Foundation Trust and the University of Bristol. We thank the patients and staff of all the hospitals who have contributed data to the National Joint Registry. We are grateful to the Healthcare Quality Improvement Partnership, the National Joint Registry Steering Committee and staff at the National Joint Registry for facilitating this work.

**Contributors** CP, AB, AJ and MW designed the study. CP, AB, AS, JMW, LH, AJ and MW reviewed the published work. CP conducted the statistical analysis and wrote the manuscript. All authors contributed to critical review of the manuscript and its revision. CP had full access to all the data and AB is the guarantor.

**Funding** This study was funded by the NIHR Biomedical Research Centre at University Hospitals Bristol NHS Foundation Trust and the University of Bristol. Adrian Sayers was supported by a MRC fellowship MR/L01226X/1.

**Disclaimer** The views expressed represent those of the authors and do not necessarily reflect those of the National Joint Registry Steering Committee or Healthcare Quality Improvement Partnership (who do not vouch for how the information is presented) or those of the NHS, the NIHR or the Department of Health and Social Care.

**Competing interests** AB and MW are involved in a separate grant to the University of Bristol funded by Stryker.

**Patient consent for publication** Not required.

**Ethics approval** Patient consent was obtained for data collection by the National Joint Registry. According to the specifications of the NHS Health Research Authority, separate informed consent and ethical approval were not required for the present study.

**Provenance and peer review** Not commissioned; externally peer reviewed.

**Data availability statement** Access to data is available from the National Joint Registry for England and Wales.

**ORCID iDs**
Chris M Penfold http://orcid.org/0000-0001-8654-353X
Adrian Sayers http://orcid.org/0000-0001-7452-5043
Michael R Whitehouse http://orcid.org/0000-0003-2436-9024

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
