## [Reviewer comments · BMJ Open]

ARTICLE DETAILS

TITLE (PROVISIONAL)	Understanding the uptake of new hip replacement implants in the UK: A cohort study using data from the National Joint Registry for England and Wales
AUTHORS	Penfold, Chris; Blom, AW; Sayers, Adrian; Wilkinson, J. Mark; Hunt, Linda; Judge, Andrew; Whitehouse, Michael R

VERSION 1 – REVIEW

REVIEWER	Fouad Chaudhry Hull and East Yorkshire Hospitals NHS Trust UK
REVIEW RETURNED	24-Feb-2019

GENERAL COMMENTS	Interesting study to look at the pattern of how new implants are taken in the UK. It would be interesting to know what was the ODEP rating of these implants when they were first used, but perhaps it was beyond the scope of this paper.
--

REVIEWER	Dr Thomas NERI Laboratory of Human Movement Biology (LIBM EA 7424), University of Lyon - Jean Monnet, France
REVIEW RETURNED	07-Mar-2019

GENERAL COMMENTS	Thank you for this interesting study. As a general comment, this article offers a very meticulous and complete data collection. This manuscript is technically well written with a strong methodology based on a large database and showing interesting epidemiologic data on THA implants used. Each step is clear and well explained. It is a well written paper. The main limitation is about the definition of new implant, which is not clear and not well explained. Moreover, it will be interesting to have more details on change, such as layer used, fixation mode (cementless?) and shape. It will be interesting to know what type of new implant is more and more used (short stem, cementless?,). More detailed comments follow: Form: Please avoid "our/we" and use the passive form Abstract: Well written. The organization corresponds to that expected by the journal. Design part: please avoid "we" and use passive form. The conclusion is supported by the results.
--

	Introduction Background is good. Please explain with more details the definition and the difference between “new” and established implants. Given it is a registry study, please try to explain how this study will change surgical practice or understanding. The objective and hypothesis of the study are clear. Methods The method section is well organized and easy to understand. Please add more details on the definition of “new and established” implants. Why did you choose 2008? It is based on something new ? (layer, fixation mode, shape..). Is this choice supported by the literature? Results: The result section is well organized. The tables are clear and understandable. Discussion: The organization of the discussion is clear and well written. The discussion highlights the strengths, the limitations of the study, and comparison with literature.
--	--

REVIEWER	Job LC van Susante Rijnstate Netherlands
REVIEW RETURNED	17-May-2019

GENERAL COMMENTS	Thank you for allowing me to review the manuscript “Understanding the uptake of new hip replacements in the UK: Analysis of data from the National Joint Registry for England and Wales”. This is the first study to describe the variation in uptake of new components by surgeons, and both surgeon and patient characteristics which may be associated with the use of new components. A cohort of 618,393 primary THAs performed by 4,979 surgeons between 2008-2017 was evaluated. Sixty-eight new cups and 72 new stems were used in 72,349 THAs (11.7%) by 2,423 surgeons (48.7%). Patients aged <55 had higher odds of receiving new rather than established implants. The authors are to be complemented with a very interesting study dealing with a very important topic. A THA is recognized to be one of the, if not the, most promising procedure to improve a patient’s quality of life. From this excellent success rate gradually younger patients are receiving these implants and as such the life time risk for revision increases. Innovations may be important to reduce revision rates. However, from the excellent results currently obtained there appears to be limited space for improvement and some recent innovations in fact turned out to increase rather than decrease the risk of revision. For this reason it is extremely important that transparency is obtained around both quantitative and qualitative consequences of newly introduced implants. Phased introduction of new medical devices together with careful monitoring has repeatedly been advocated to avoid hyped introduction with sometimes dreadful consequences in case of unforeseen disadvantages of an innovation.
--

The current manuscript clearly reveals that a rather large number of new implant components for THA were introduced in England and Wales between 2008-2017. I have a few comments:

Abstract

- Page 4, line 38-40. This sentence is not easy to understand. Assume surgeons performed a median of 22 THR per year, please clarify. The 5.0% and 9.4% are also difficult to understand, please provide units to which these percentages refer.
- A median of 22 THR a year appears to be very low. The range from 1-3938 is extremely wide. Maybe it would be interesting to comment on these numbers further on in the manuscript. Should there be a minimum of implants performed a year?

Background

- Page 6, line 12-14. "Such patients may benefit the most....". You may also add a remark that at the same time these patients are also at risk for exposure to the uncertainty around hyped introduction of new and unestablished implants.

Methods

- Line 18, "We excluded patients who received a resurfacing rather than stemmed THR". Why were resurfacing THRs excluded? These were new implants where the hyped introduction typically did not follow the IDEAL recommendations (10).

Results

- Line 30. "performed a median of 22 THRs over the period". Please clarify whether this was 22/year or over the entire period. In case a median of 22 THRs were implanted over the entire period I would suggest to incorporate a section in methods where this information is provided. Together with transparency on the number of new components introduced the manuscript would improve if transparency is also provided on how many surgeons have placed from example less than 25 THR a year.
- In general I have the feeling in its current form the manuscript does provide interesting data and numbers however the message may become stronger when the authors decide to speculate a bit on the direction they believe it should go. In other words should we stimulate the introduction of more new components or temper it? Should we encourage surgeons who implant less than for example 10 THR / year to proceed? Is there a correlation between these two?
- Page 10, line 15. "a third of the new stems and cups have been used in <10 THR". In table S1 I counted 29 new cups which were implanted <5 times, whereas for stems in table S2 this were 39 stems implanted < 5 times. This is interesting information available in the manuscript. I would suggest to add a subsection providing an overview of the number of surgeons implanting less than 25 THR / year and the number of newly introduced stems and cups implanted less than 5 times. This information may subsequently be discussed in the discussion section and the authors are invited to speculate on whether this should be stimulated / accepted / controlled.

Discussion

- This may be my personal preference however I would suggest to discuss strengths and limitations further at the end of the discussion.

	- The authors present a large variation on numbers of newly introduced components between the countries Australia, UK and Finland. Maybe they could go a bit further than just presenting these data. Are they critical on the number in the UK versus Australia or vice versa? Do they think the IDEAL framework should be followed better than is happening at the moment? - Page 12-13 line 3-4. "The volume of THRs performed by surgeons using new components was often low". Please comment, is this a concern? To me it is! Should we control this? In general this is a very interesting paper with important information. In its current form it is a bit of a summary of numbers. Maybe the manuscript can gain impact or interest when the authors incorporate their view on the data and introduce a dot on the horizon where they think we should go with monitoring the introduction of new components. Should we stimulate or temper the introduction of new implants? What to think about surgeons who perform <25 THR per year, should they proceed?
--	---

REVIEWER	Caren Rose University of British Columbia, Canada
REVIEW RETURNED	10-Jul-2019

GENERAL COMMENTS	This study by Penfold et al describes the uptake of new medical device components for total hip replacements in the UK, as well as the characteristics of surgeons who use new components and patients who receive them. Some new components had high-profile failures, and implementation of the IDEAL framework for phased introduction of new medical devices is suggested. Overall, I found the objectives to be conflicting. This paper is descriptive in nature, and there is value in the data on uptake, but the discussion needs to flesh out in more detail how the information from this paper can be used. Given the data shown, it is hard to determine whether the rate of uptake of these new devices is appropriate or not. Major Comments:  • The main purpose of the study is not clear. If it is to describe the uptake of the new components this needs to gel with objectives 3 and 4. I have suggested ways to adjust the models that would keep them in line with the uptake message of the paper. • Please describe the rationale for choosing Jan 1 2008 as the cutoff for new vs established use. • Figure 2 is the MOST important figure in the manuscript. Describing this in more detail in the results is needed. I would put this as Figure 1. • MODELING: The exclusion of 30% of subjects from multivariable analyses due to missing BMI is huge red flag, and makes the results of these analyses in isolation uninformative. Missing BMI is very unlikely to be at random. Perhaps patients with higher BMI are more difficult to weigh and therefore patients with missing BMI might have higher BMI. To fix:  1. Compare population with BMI and without BMI to identify any major differences. 2. Run a sensitivity analyses on the full population, excluding BMI as a factor. 3. Run an analysis of the full population, adding missing BMI as a dummy variable.
--

4. Compare the above analyses to the original analyses and use the consistencies or differences to make further interpretation.

- In order to align the first 2 objectives about uptake with the surgeon and patient level modeling, I think your outcome of interest in the logistic regression models needs to be surgeons who used a new device within X years (e.g. 3 years) of introduction, or patients who had a THR with a device that was within X years of introduction. This will help you identify which patients/surgeons use these new devices early. Otherwise, policies within hospitals, insurance companies might change the type of surgeon/patient that use devices, and as devices are around for longer than X years the use of them is no longer related to uptake. If for example, one device appears to have lower early revision rates, it may be picked up more ubiquitously and the addition of the late surgeries to your model are no longer informative, and may give different results.

Minor Comments:

Introduction:

- Can you elaborate on the recommendations from IDEAL. The discussion and results focus on 'rapid' uptake among other things that would benefit from some accessible knowledge (i.e. in this paper), about what the key standards are.
- Can you please make the following sentence more clear: There is wide variation between and within regions of common surgical procedures.[i.e. perhaps that it is variation in the 'types' of surgeries or components used, the type of patients surgeries performed on.]

Methods:

- Figure 1 can be in black, greys and white, colour doesn't add much. What is the key take-home from Figure 1? The increased use of newer components is self-evident given the time of introduction. In the text you report results from 2016 in isolation. Why did you choose this timepoint? Perhaps you could remove Figure 1 and mention in text that the proportion of 'new' components used leveled out around 2014-108 at X%.
- Be more explicit about how you chose the cutoffs for the variables in the models. For example, why did you choose age=55/80 as cutoffs for age and why did you choose <10% young vs >10% young for surgeon cutoffs? Were these percentiles from the data, are these clinically relevant for some reason?
- Is the variable for number of stems and cups too highly related to use of a new stem/cup? I am concerned that these are collinear and this variable should be removed from the model.
- Given the distribution of the number of THRs performed per year by each surgeon, it might make more sense to show this variable by the quartile cutoffs instead of per 10 additional cases.
- Can you also include centre or hospital as a hierarchical level in your model? I would imagine that use of devices is correlated within and institution.

Results:

- IQR- the interquartile range is one number..you want Q1 and Q3 or p25 and p75 to represent the first and third quartiles when you present medians and quartiles. Your numbers are ok, but need to change language of IQR.
- There are 68 new cups and 72 new stems..how many established cups and stems are there?

	 • Can you define the use of 'rapid'. Used to describe 5000 total uses in 1000 days, hard to know if this is indeed rapid given the number of total surgeries. Perhaps you can remove qualifier and just state days. • When commenting on days to use of 5000 cups use the actual days. i.e. 947 instead of <1000 or 1635 instead of approximately 1500 • For top 5 used brands can you comment on 1,3,5 year revision rates? If you want to comment on the quick uptake of these components, and whether it is appropriate or not if would be helpful to have this info. For example. If the top new brand for which 5000 cups was used in 1000 days, has lower 1 year revision rates for 'similar' patients, perhaps the uptake should be more rapid. • Multivariable adjusted is redundant • Your overall description of the population is it among all patients? Or only among the ones included in the final study? It is a bit misleading to be inconsistent here. As you are excluding 30% of the population in the final analysis..which may not be representative of the overall population you are including.
--	---

VERSION 1 – AUTHOR RESPONSE

Reviewer: 1

Reviewer Name: Fouad Chaudhry

Institution and Country: Hull and East Yorkshire Hospitals NHS Trust, UK Please state any competing interests or state 'None declared': None declared

Interesting study to look at the pattern of how new implants are taken in the UK. It would be interesting to know what was the ODEP rating of these implants when they were first used, but perhaps it was beyond the scope of this paper.

We agree that the ODEP rating of these implants at the time of their first use may be interesting. Unfortunately it is not currently possible to use historic ODEP ratings.

Reviewer: 2

Reviewer Name: Dr Thomas NERI

Institution and Country: Laboratory of Human Movement Biology (LIBM EA 7424), University of Lyon - Jean Monnet, France Please state any competing interests or state 'None declared': none declared

Thank you for this interesting study.

As a general comment, this article offers a very meticulous and complete data collection. This manuscript is technically well written with a strong methodology based on a large database and showing interesting epidemiologic data on THA implants used. Each step is clear and well explained. It is a well written paper.

The main limitation is about the definition of new implant, which is not clear and not well explained. Moreover, it will be interesting to have more details on change, such as layer used, fixation mode (cementless?) and shape. It will be interesting to know what type of new implant is more and more used (short stem, cementless?,).

We have clarified our definition of new implant components in the Methods. In response to Reviewer 4 we have also incorporated a 5-year time window during which implant components were classified as 'new' and after this period they became 'established'.

We did not consider the extent to which 'new' implant components changed from previous components. This limitation is included in the discussion, where we acknowledge that new components may be minor modifications of existing components, but also that a single implant component may cover multiple variations.

We agree that trends in the types of implants being introduced (e.g. fixation method or stem length) may be of interest but feel that it is outside the scope of this study.

More detailed comments follow:

Form: Please avoid "our/we" and use the passive form

We have respectfully chosen to leave the article written in the active form. We hope that the reviewer will permit our preference for this rather than the passive form.

Abstract:

Well written. The organization corresponds to that expected by the journal.

Design part: please avoid "we" and use passive form.

Please see our response above.

The conclusion is supported by the results.

Introduction

Background is good. Please explain with more details the definition and the difference between "new" and established implants.

Please see our response above.

Given it is a registry study, please try to explain how this study will change surgical practice or understanding.

We have included an additional paragraph at the end of the Discussion in which we detail how our findings relate to and influence wider discussions about the phased introduction of new implant components (Page 16, line 33).

The objective and hypothesis of the study are clear.

Methods

The method section is well organized and easy to understand.

Please add more details on the definition of “new and established” implants. Why did you choose 2008? It is based on something new ? (layer, fixation mode, shape..). Is this choice supported by the literature?

Please see our response above with respect to more detail on the definition of ‘new’ and ‘established’ implant components. We have expanded our reasoning for these definitions, but there is not any previous literature on the uptake of new joint replacement implants to use as a starting point for these definitions.

Results:

The result section is well organized.

The tables are clear and understandable.

Discussion:

The organization of the discussion is clear and well written. The discussion highlights the strengths, the limitations of the study, and comparison with literature.

Reviewer: 3

Reviewer Name: Job LC van Susante

Institution and Country: Rijnstate, Netherlands Please state any competing interests or state ‘None declared’: None declared

Thank you for allowing me to review the manuscript “Understanding the uptake of new hip replacements in the UK: Analysis of data from the National Joint Registry for England and Wales”.

This is the first study to describe the variation in uptake of new components by surgeons, and both surgeon and patient characteristics which may be associated with the use of new components. A cohort of 618,393 primary THAs performed by 4,979 surgeons between 2008-2017 was evaluated. Sixty-eight new cups and 72 new stems were used in 72,349 THAs (11.7%) by 2,423 surgeons (48.7%). Patients aged <55 had higher odds of receiving new rather than established implants.

The authors are to be complemented with a very interesting study dealing with a very important topic. A THA is recognized to be one of the, if not the, most promising procedure to improve a patient's quality of life. From this excellent success rate gradually younger patients are receiving these implants and as such the life time risk for revision increases. Innovations may be important to reduce revision rates. However, from the excellent results currently obtained there appears to be limited space for improvement and some recent innovations in fact turned out to increase rather than decrease the risk of revision. For this reason it is extremely important that transparency is obtained around both quantitative and qualitative consequences of newly introduced implants. Phased introduction of new medical devices together with careful monitoring has repeatedly been advocated to avoid hyped introduction with sometimes dreadful consequences in case of unforeseen disadvantages of an innovation.

The current manuscript clearly reveals that a rather large number of new implant components for THA were introduced in England and Wales between 2008-2017. I have a few comments:

Abstract

- Page 4, line 38-40. This sentence is not easy to understand. Assume surgeons performed a median of 22 THR per year, please clarify. The 5.0% and 9.4% are also difficult to understand, please provide units to which these percentages refer.

We have split this sentence into 2 sentences and revised the wording as follows:

“Surgeons performed a median total of 22 THRs (25%-75%=5-124, range 1-3,938) in the period 2008-2017. Surgeons used new stems in a median of 5.0% (25%-75%=1.3-16.1%) and new cups in a median of 9.4% (25%-75%=2.8-26.7%) of their THRs.”

- A median of 22 THR a year appears to be very low. The range from 1-3938 is extremely wide. Maybe it would be interesting to comment on these numbers further on in the manuscript. Should there be a minimum of implants performed a year?

We thank the reviewer for this comment. We have clarified that the comparatively low volume of THRs is due to come surgeons starting part way through this period, retiring or changing their practice. For further clarity, in the Results in the main text we have also included the median number of THRs surgeons performed per year, excluding calendar-years in which a surgeon performed no THRs.

However, it was outside the scope of this study to consider the outcomes of THRs performed by surgeons with differing annual THR volumes. Consideration of the effect of surgeon volume on outcomes and any recommendations about annual volume is an

interesting and complex topic which would be best addressed in a dedicated study. Our group has carried out separate research in this area which we hope will be published soon.

Background

- Page 6, line 12-14. "Such patients may benefit the most...". You may also add a remark that at the same time these patients are also at risk for exposure to the uncertainty around hyped introduction of new and unestablished implants.

We have added the sentence 'However, they may also be affected for the longest time if these developments lead to poorer outcomes.' (page 7, line 15).

Methods

- Line 18, "We excluded patients who received a resurfacing rather than stemmed THR". Why were resurfacing THRs excluded? These were new implants where the hyped introduction typically did not follow the IDEAL recommendations (10).

We excluded patients who received a resurfacing THR since the patients who receive these implants are a very different demographic than those receiving stemmed THRs (significantly younger and much more likely to be male). The annual volume of resurfacing THRs is also now very low (~550 in 2017) and decreasing further. Therefore trends in the uptake of resurfacing implants are likely to be limited in their generalisability.

We have included the following text in the Methods section (page 8, line 30):

"Resurfacing THRs were excluded since patients who receive these are a very different demographic from those receiving stemmed THRs (significantly younger and more likely to be male), and the annual volume is very low (~550 in 2017) and decreasing [14]."

Results

- Line 30. "performed a median of 22 THRs over the period". Please clarify whether this was 22/year or over the entire period. In case a median of 22 THRs were implanted over the entire period it would suggest to incorporate a section in methods where this information is provided. Together with transparency on the number of new components introduced the manuscript would improve if transparency is also provided on how many surgeons have placed from example less than 25 THR a year.

We have changed the description of the median volume of THRs performed to '...a median total of 22 THRs between 2008 and 2017...'. We have also included the number of surgeon calendar-years in which <10 THRs were performed. We have opted for a lower threshold of <10 THRs/year rather than the reviewer's suggestion of <25 THRs/year to maintain consistency with our surgeon-level sensitivity analyses.

- In general I have the feeling in its current form the manuscript does provide interesting data and numbers however the message may become stronger when the authors decide to speculate a bit on the direction they believe it should go. In other words should we stimulate the introduction of more new components or temper it? Should we encourage surgeons who implant less than for example 10 THR / year to proceed? Is there a correlation between these two?

We have included an additional paragraph at the end of the Discussion in which we detail how our findings relate to and influence wider discussions about the phased introduction of new implant components (page 16, line 33).

- Page 10, line 15. "a third of the new stems and cups have been used in <10 THR". In table S1 I counted 29 new cups which were implanted <5 times, whereas for stems in table S2 this were 39 stems implanted < 5 times. This is interesting information available in the manuscript. I would suggest to add a subsection providing an overview of the number of surgeons implanting less than 25 THR / year and the number of newly introduced stems and cups implanted less than 5 times. This information may subsequently be discussed in the discussion section and the authors are invited to speculate on whether this should be stimulated / accepted / controlled.

We thank the reviewer for their comment regarding the number of implant components used in very few cases. The reviewer may have been mistaken regarding the number of new stems and cups implanted <5 times. However, given the revised definition of 'new' implant components we have updated the results in this section.

We have included an overview of the number of surgeons implanting less than 10 THRs/year (page 11, line 32). This maintains consistency with our sensitivity analyses in which we exclude these very low volume surgeons.

Discussion

- This may be my personal preference however I would suggest to discuss strengths and limitations further at the end of the discussion.

We have respectfully chosen to leave the discussion of strengths and weaknesses early in the Discussion. We hope that the reviewer will permit our preference for this rather than moving it towards the end of the Discussion as we feel it does influence the interpretation of the rest of the discussion and make clear the limitations of the data presented that need to be borne in mind when considering the speculative elements added to the discussion as suggested by the reviewer.

- The authors present a large variation on numbers of newly introduced components between the countries Australia, UK and Finland. Maybe they could go a bit further than just presenting these data. Are they critical on the number in the UK versus Australia or vice versa? Do they think the IDEAL framework should be followed better than is happening at the moment?

Our study describes the rate of introduction of new implant components. However, this study does not provide insight into whether this rate is in itself a good or bad thing. We have expanded our discussion to consider how a healthcare system which supported graduated introduction of new implants would naturally limit the rate of introduction. Specifically, we have added the following sentence (page 15, line 16):

“It is unclear whether 16 new implant components/year is of itself a good or bad thing. However, a healthcare system which supported a graduated introduction of new components, where the use of new components is restricted to specialised centres,[18] would provide a natural limit on the rate of introduction of new components until satisfactory and robust evidence is generated to support their more widespread use.”

- Page 12-13 line 3-4. “The volume of THRs performed by surgeons using new components was often low”. Please comment, is this a concern? To me it is! Should we control this?

Our study does not directly consider whether surgeons who use implants only in low volumes, either due to low overall volume or because they switch frequently, should be monitored. We agree that this is an interesting question but it was outside the scope of this study. However, we have already discussed and referenced our findings in context with other studies which have found elevated revision rates for surgeons who switch implants frequently and for the early THRs performed after a surgeon has switched.

In general this is a very interesting paper with important information. In its current form it is a bit of a summary of numbers. Maybe the manuscript can gain impact or interest when the authors incorporate their view on the data and introduce a dot on the horizon where they think we should go with monitoring the introduction of new components. Should we stimulate or temper the introduction of new implants? What to think about surgeons who perform <25 THR per year, should they proceed?

As discussed above, we have included an additional paragraph at the end of the Discussion in which we detail how our findings relate to and influence wider discussions about the phased introduction of new implant components. We have also included an additional paragraph in which we discuss directions for future research, largely focussing on broadening our understanding of surgeon-level factors associated with risk of revision after THR (page 16, line 47).

Reviewer: 4

Reviewer Name: Caren Rose

Institution and Country: University of British Columbia, Canada Please state any competing interests or state 'None declared': None declared

This study by Penfold et al describes the uptake of new medical device components for total hip replacements in the UK, as well as the characteristics of surgeons who use new components and patients who receive them. Some new components had high-profile failures, and implementation of the IDEAL framework for phased introduction of new medical devices is suggested.

Overall, I found the objectives to be conflicting. This paper is descriptive in nature, and there is value in the data on uptake, but the discussion needs to flesh out in more detail how the information from this paper can be used. Given the data shown, it is hard to determine whether the rate of uptake of these new devices is appropriate or not.

Major Comments:

- The main purpose of the study is not clear. If it is to describe the uptake of the new components this needs to gel with objectives 3 and 4. I have suggested ways to adjust the models that would keep them in line with the uptake message of the paper.

Thank you for these helpful suggestions. We have incorporated them into our study as detailed in our responses to later comments.

- Please describe the rationale for choosing Jan 1 2008 as the cutoff for new vs established use.

We have included more detail about our choice of Jan 1st 2008 as our earliest date for an implant component to be considered 'new', and have incorporated your suggestion of a time window after which a 'new' component was classified as 'established' (page 8, line 52).

- Figure 2 is the MOST important figure in the manuscript. Describing this in more detail in the results is needed. I would put this as Figure 1.

We have changed this to Figure 1 as suggested. Regarding the suggestion to describe this in more detail, we have included additional details in response to comments below. We would be happy to provide further detail if necessary.

- MODELING:

The exclusion of 30% of subjects from multivariable analyses due to missing BMI is huge red flag, and makes the results of these analyses in isolation uninformative. Missing BMI is very unlikely to be at random. Perhaps patients with higher BMI are more difficult to weigh and therefore patients with missing BMI might have higher BMI. To fix:

1. Compare population with BMI and without BMI to identify any major differences.
2. Run a sensitivity analyses on the full population, excluding BMI as a factor.
3. Run an analysis of the full population, adding missing BMI as a dummy variable.
4. Compare the above analyses to the original analyses and use the consistencies or differences to make further interpretation.

One of the key reasons BMI data were missing from the NJR in the early years of data collection was because it was not included on the earliest version of the data collection forms. Later iterations of the data collection forms included BMI.

We thank the reviewer for their suggestions regarding further analysis to determine the potential impact of excluding cases with missing BMI data on our findings. We have added analysis suggestions 1, 2 and 4 as sensitivity analyses, with associated Methods described, and commentary in the Results and Discussion. We have not undertaken the suggestion to include a dummy variable for missing BMI (suggestion 3) since, in a nonrandomised study, this approach would normally result in biased

estimates - Groenwold RHH et al. *'Missing covariate data in clinical research: when and when not to use the missing-indicator method for analysis'*. CMAJ. 2012

The results of these analyses indicate that there were no important differences between patients with and without BMI for all variables apart from BMI. We are not able to comment on whether the missing BMI values themselves are missing at random or missing not at random. Much as patients with a higher BMI may be more difficult to weight, it is equally possible that surgeons may be more likely to record the BMI when this has represented an intraoperative challenge so there may be many factors at play.

- In order to align the first 2 objectives about uptake with the surgeon and patient level modeling, I think your outcome of interest in the logistic regression models needs to be surgeons who used a new device within X years (e.g. 3 years) of introduction, or patients who had a THR with a device that was within X years of introduction. This will help you identify which patients/surgeons use these new devices early. Otherwise, policies within hospitals, insurance companies might change the type of surgeon/patient that use devices, and as devices are around for longer than X years the use of them is no longer related to uptake. If for example, one device appears to have lower early revision rates, it may be picked up more ubiquitously and the addition of the late surgeries to your model are no longer informative, and may give different results.

We have included this helpful suggestion of a time window, after which a 'new' implant component would be considered 'established'. We felt that a 5-year rather than 3-year time window would allow 'new' implant components with slower uptake to disseminate more widely into practice but the 5-year outcomes of those with rapid uptake would not yet be available.

We have updated the Methods to reflect this revised classification of 'new' and 'established' implant components (page 8, line 47). We have also updated the Results and Discussion sections as a consequence of this revision.

Minor Comments:

Introduction:

- Can you elaborate on the recommendations from IDEAL. The discussion and results focus on 'rapid' uptake among other things that would benefit from some accessible knowledge (i.e. in this paper), about what the key standards are.

We have included more detail about the IDEAL recommendations in the context of the ASR hip replacements. We have modified the Introduction with the following sentences (page 7, line 29):

"The rapid uptake of ASR hip replacements before the publication of supporting evidence bypassed IDEAL Stages 2a ('Development') and 2b ('Early dispersion and exploration'). Instead, long-term monitoring was relied on to monitor outcomes (Stage 4). It is not clear whether the uptake of newer implants has also been rapid."

- Can you please make the following sentence more clear: There is wide variation between and within regions of common surgical procedures.[i.e. perhaps that it is variation in the 'types' of surgeries or components used, the type of patients surgeries performed on.]

We have revised this sentence as follows (page 7, line 36):

“There is wide variation between and within regions in the use of common surgical procedures, which are only explained to a small degree by differing patient demands and diagnostic practices.”

Methods:

- Figure 1 can be in black, greys and white, colour doesn't add much. What is the key take-home from Figure 1? The increased use of newer components is self-evident given the time of introduction. In the text you report results from 2016 in isolation. Why did you choose this timepoint? Perhaps you could remove Figure 1 and mention in text that the proportion of 'new' components used leveled out around 2014-108 at X%.

We have removed this Figure and removed the associated text.

- Be more explicit about how you chose the cutoffs for the variables in the models. For example, why did you choose age=55/80 as cutoffs for age and why did you choose <10% young vs >10% young for surgeon cutoffs? Were these percentiles from the data, are these clinically relevant for some reason?

We have added additional detail to our Methods in order to be more explicit about the cutoffs for the variables in the models (page 9, line 23 and page 9, line 39).

- Is the variable for number of stems and cups too highly related to use of a new stem/cup? I am concerned that these are collinear and this variable should be removed from the model.

Prompted by this concern we have checked the variance inflation factors for all our regression models and found all VIFs <2.5. This suggests that multicollinearity was unlikely to have inflated the variance of any coefficients in our models. We have therefore kept the number of stem-cup combinations in our regression models.

- Given the distribution of the number of THRs performed per year by each surgeon, it might make more sense to show this variable by the quartile cutoffs instead of per 10 additional cases.

Thank you for this suggestion. However, we feel that keeping this exposure as 'per 10 additional cases' will be more interpretable by readers.

- Can you also include centre or hospital as a hierarchical level in your model? I would imagine that use of devices is correlated within and institution.

Unfortunately we are not able to include centre or hospital data alongside patient and surgeon-level data as we were not permitted to use the data in this way due to the risk of deanonymisation of the data.

Results:

- IQR- the interquartile range is one number..you want Q1 and Q3 or p25 and p75 to represent the first and third quartiles when you present medians and quartiles. Your numbers are ok, but need to change language of IQR.

We have replaced 'IQR' with '25%-75%'.

- There are 68 new cups and 72 new stems..how many established cups and stems are there?

In the first paragraph of the 'Results' section, under the heading 'Overall use of implant components' we report the total number of different stems, different cups and different stem-cup combinations used between 2008-2017 (page 11, line 24).

- Can you define the use of 'rapid'. Used to describe 5000 total uses in 1000 days, hard to know if this is indeed rapid given the number of total surgeries. Perhaps you can remove qualifier and just state days.
- When commenting on days to use of 5000 cups use the actual days. i.e. 947 instead of <1000 or 1635 instead of approximately 1500

We have updated the descriptive results to include the actual number of days.

- For top 5 used brands can you comment on 1,3,5 year revision rates? If you want to comment on the quick uptake of these components, and whether it is appropriate or not if would be helpful to have this info. For example. If the top new brand for which 5000 cups was used in 1000 days, has lower 1 year revision rates for 'similar' patients, perhaps the uptake should be more rapid.

We thank the reviewer for this suggestion, however the revision rates of individual implant components overlooks the significant variation in revision rates between constructs (combination of stem, cup and bearing surface) with at least one common component and between different age/gender substrata – please see:

Deere KC, Whitehouse MR, Porter M, Blom AW, Sayers A. Assessing the non-inferiority of prosthesis constructs used in hip replacement using data from the National Joint Registry of England, Wales, Northern Ireland and the Isle of Man: a benchmarking study. BMJ Open. 2019 Apr 1;9(4):e026685..

We feel that it would be more appropriate for surgeons to consider the revision rates of new implant components within the constructs which are relevant to their current practice and patients they are treating, rather than revision rates which do not account for the other construct components.

- Multivariable adjusted is redundant

We have removed 'adjusted', leaving only 'multivariable'.

- Your overall description of the population is it among all patients? Or only among the ones included in the final study? It is a bit misleading to be inconsistent here. As you are excluding 30% of the population in the final analysis..which may not be representative of the overall population you are including.

The overall description of the population is for all eligible patients as defined in the 'Study sample' section of the Methods. As discussed in relation to the suggested sensitivity analyses (above), the subset of patients with complete data had very similar characteristics to those with incomplete data. We are therefore confident that they are unlikely to be a biased subset.

VERSION 2 – REVIEW

REVIEWER	Job LC van Susante Rijnstate Arnhem The Netherlands
REVIEW RETURNED	05-Sep-2019
GENERAL COMMENTS	The authors dealt appropriately with the reviewer's comments